# Vision Feedback Control for the Automation of the Pick-and-Place of a Capillary Force Gripper

**DOI:** 10.3390/mi13081270

**Published:** 2022-08-07

**Authors:** Takatoshi Ito, Eri Fukuchi, Kenta Tanaka, Yuki Nishiyama, Naoto Watanabe, Ohmi Fuchiwaki

**Affiliations:** Department of Mechanical Engineering, Yokohama National University, Yokohama 240-8501, Japan

**Keywords:** micromanipulation, capillary force, water, vision feedback, non-contact

## Abstract

In this paper, we describe a newly developed vision feedback method for improving the placement accuracy and success rate of a single nozzle capillary force gripper. The capillary force gripper was developed for the pick-and-place of mm-sized objects. The gripper picks up an object by contacting the top surface of the object with a droplet formed on its nozzle and places the object by contacting the bottom surface of the object with a droplet previously applied to the place surface. To improve the placement accuracy, we developed a vision feedback system combined with two cameras. First, a side camera was installed to capture images of the object and nozzle from the side. Second, from the captured images, the contour of the pre-applied droplet for placement and the contour of the object picked up by the nozzle were detected. Lastly, from the detected contours, the distance between the top surface of the droplet for object release and the bottom surface of the object was measured to determine the appropriate amount of nozzle descent. Through the experiments, we verified that the size matching effect worked reasonably well; the average placement error minimizes when the size of the cross-section of the objects is closer to that of the nozzle. We attributed this result to the self-alignment effect. We also confirmed that we could control the attitude of the object when we matched the shape of the nozzle to that of the sample. These results support the feasibility of the developed vision feedback system, which uses the capillary force gripper for heterogeneous and complex-shaped micro-objects in flexible electronics, micro-electro-mechanical systems (MEMS), soft robotics, soft matter, and biomedical fields.

## 1. Introduction

Electronics are becoming increasingly sophisticated while decreasing in size. For high-resolution displays, 0201 sized electronic chip parts and micro-LEDs of less than 100 μm have been developed [1]. Microfiber assembly is also one of the most significant technologies for the advancement of information processing instruments [2].

In conventional surface mounting technology (SMT), the suction force generated by compressed air is used mainly to pick up flat components. However, as the size of the chip decreases, a higher mounting speed and accuracy are required to achieve an adequate mounting density. Transfer printing technologies are feasible methods for assembling micro-LEDs [3,4,5]. Mechanical grippers have important applications, such as medical operations on eyeballs [6]. Shape memory polymers have also been investigated as a feasible micromanipulation method [7,8].

These techniques are based on the contact between the micro-object and the gripper, which may damage the targets. In contact-type grippers, it is necessary to control the gripping force while handling fragile objects. A noncontact-type gripper decreases the possibility of damaging the parts being handled.

In liquids, various non-contact micromanipulation methods have been reported, such as using steady streaming [9,10], electric fields [11], and laser tweezers [12,13] for fragile positioning, such as biological cells and microorganisms.

In the atmosphere, ultrasonic levitation [14,15] and a liquid bridge force (hereinafter referred to as capillary force-based gripper) are categorized as non-contact grasping methods with self-alignment [16,17]. Several studies have been conducted on capillary force grippers for pick-and-place operations [18,19,20,21,22]. Estimation of capillary force [23,24], and image processing technologies [25,26] are also significant research categories for the automation of micromanipulation and the classification of micro fossils [25], micro particles [26], soft robotics [27], soft matters [28], and biomedical fields.

In a previous article, Tanaka et al. reported a double nozzle gripper using two liquid transporting methods: a diaphragm pump and the capillary phenomenon [22]. This study focused on the picking and placing of 1-mm sized objects with various shapes and did not describe a vision feedback control method for decreasing the positioning errors.

In this paper, we describe the details of the image processing procedure and the investigation of a size matching effect that influences final alignment accuracy [28]. We also describe the attitude control by a shape matching effect between the nozzle shape and the picked object [28,29].

The remainder of the paper is structured as follows. We describe the static analysis in Section 2; the mechanical design in Section 3; the experimental setup, as shown in Figure 1, in Section 4; the image processing in Section 5; the success rate, placing error, and attitude control in Section 6; and concluding remarks and future research scope in Section 7. (See also Appendix A “Digest movie” for the outline).

## 2. Approximation of the Capillary Force

In this section, we estimate the capillary forces using the geometrical parameters of the objects when the top surface of the picked-up object is a plane.

Figure 2 shows the capillary bridge between the nozzle and the object. Here, the shape of the bottom of the liquid bridge is approximated to be circular. We defined the radius of the interface between the liquid bridge and the nozzle as *r*_2_ and the radius of the interface between the liquid bridge and the object as *r*_1_. In this case, assuming that the water spreads over the nozzle, we could estimate *r*_2_ to be the radius of the nozzle. *h* is the height of the liquid bridge, as shown in Figure 2. $\;R_{1}$ and $R_{2}$ are the arc-approximated meniscus radii of the liquid bridges caused by the interfacial tension, respectively. In this case, it is the length of the wispiest part of the liquid bridge. Moreover, $\theta_{1}$ and $\theta_{2}$ are the contact angles between the object and the liquid and the nozzle and the liquid, respectively.

The capillary force is determined geometrically if the section curves of the meniscus are approximated as parts of the arc [23,24]. In this case,$\;R_{1}$ and $R_{2}$ are expressed as follows:(1)$$R_{1}=\frac{h}{\cos\theta_{1}+\cos\theta_{2}}$$
(2)$$R_{2}=r_{2}-R_{1}\left(1-\sin\theta_{2}\right)$$

The mean curvature *C* of the meniscus is determined as follows:(3)$$C=\frac{1}{R_{2}}-\frac{1}{R_{1}},$$
where $R_{2}$ is positive. When the shape is convex, $R_{1}$ is defined as negative. The difference in pressure across the interface, given by Laplace’s equation, is as follows:(4)Δp=γC

Here, γ is the surface tension, and $r_{1}$ is given as follows:(5)$$r_{1}=r_{2}+R_{1}\left(\sin\theta_{2}-\sin\theta_{1}\right).$$

Thus, the suction force acting on the flat plane (top surface of the cube) by the Laplace pressure $F_{L}$ is expressed as follows:(6)$$F_{L}=\pi r_{1}^{2}\Delta p.$$

The pulling force acting on the flat plane owing to the surface tension is given as follows:(7)$$F_{T}=2\pi r_{1}\gamma \sin\theta_{1}.$$

From (6) and (7), the required capillary force $F_{C}$, which is the sum of $F_{L}$ and $F_{T}$, is given as follows:(8)$$F_{C}=F_{L}+F_{T}\;=\pi \gamma r_{1}\left[r_{1}\left(\frac{1}{R_{2}}-\frac{1}{R_{1}}\right)+2\sin\theta_{1}\right].$$

## 3. Design

The gripper consists of a flow device, flow channel, tank, shaft, and valve stem, all fabricated from acrylic. The flow channel was made of a silicon tube. A stainless-steel tube was used at the end of the flow channel, which is called the nozzle. We used the valve stem and the shaft to refill water and form droplets, and we adopted a spring interlocking mechanism, developed by Hagiwara et al. [21], to move them with the Z-stage.

Figure 3 illustrates the filling of water in the channel through capillary action. As shown in the left panel, even if the water at the tip of the channel disappears owing to evaporation or experiment conduction, water can again be filled to the tip of the channel by pulling down the valve and opening the channel, as shown in the right panel.

Figure 4 shows the formation of a droplet. When the water channel moves down, the valve closes the channel to prevent backflow (left panel), and the shaft then pushes the channel and forms the droplet at the channel tip using a diaphragm pump mechanism. Since the deformed volume of the diaphragm and the volume displaced from the shaft are almost equal, the volume of the droplet is determined by measuring the length of the shaft pushed in.

The object is picked up by contacting a formed droplet with the object. The object is placed by first contacting and applying a formed droplet to the place surface, and then contacting the bottom surface of the object picked up by the nozzle with the applied droplet. Table 1 shows the parameters of the micro-objects used in the pick-and-place experiments.

In the experiments, we used various sizes and shapes of nozzles for each object. The experimental conditions and parameters used to calculate the estimated value of the capillary force for each condition are organized and shown in Table 2; the estimated capillary forces are also shown. h, $\theta_{1}$, and $\theta_{2}$ were typical values measured from the charge coupled device (CCD) camera image during the experiments. The surface tension *γ*, measured by the ring method, was 72.9 mN/m; we used purified water as the liquid.

From Table 2, it can be seen that the estimated capillary force had a much larger value than the corresponding gravitational force, and hence, this gripper could grasp the object under all conditions if there were no additional adhesive or electrostatic forces.

## 4. Experimental Setup

To increase the success rate of the pick-and-place, we developed an automatic pick-and-place with vision feedback function. The organization of the entire setup is shown in Figure 5.

In this study, the work surface was positioned in the X- and Y-axis with a positioning resolution of 1 μm and a repeatability of ±0.5 μm (YA10A-L1, Kohzu Precision Co., Ltd., Kawasaki City, Japan). The gripper was connected to a linear stage (MMU-40X, Chuo Precision Industrial Co., Ltd., Tokyo, Japan) and moved along the *Z*-axis. It was also connected to the interlocking mechanism, which controls the shaft and the valve to form a droplet. The traverse limits of the XY- and Z-stages were ±50 mm and ±5 mm, respectively.

To measure the position of the samples, the vision feedback system by OpenCV was used. This system imported the image from the CCD camera set on top of the work bench. The camera had a resolution of 2432 × 2050 pixels (CV-H500M, KEYENCE Corp., Osaka, Japan). The CCD camera also attached to the side of the workbench, and it had a resolution of 1216 × 1025 pixels (CV-200C, KEYENCE Corp.). Each camera had a magnification lens (LA-LM510, Keyence Co.), and the depth of field was 1.28 mm.

The pixel resolutions of the top and side cameras were, approximately, 7 μm/pixel and 9 μm/pixel, respectively. We confirmed that the image processing and gravity center detection process in the vision feedback resulted in a final *X*- and *Y*-axis measurement resolution of 1 µm because of the averaging effect.

LabVIEW2018 (National Instruments, Austin, TX, USA) was used as the programming language, and OpenCV was used to conduct image processing. The detailed procedure for the image processing is described in Section 5. The pick-and-place process is as follows:(1)The object is placed within view of the top camera on the surface to measure the X- and Y- coordinates;(2)The XY-stage is moved so that the placement position on the work surface is directly below the nozzle;(3)The gripper is moved up and down by the Z-stage, and a droplet is applied to the work surface;(4)The XY-stage is moved, and the object is placed directly under the nozzle;(5)The gripper moves first down and then up to pick up the object;(6)The XY-stage is moved such that the target placement position is directly below the nozzle grasping the object;(7)The gripper moves down to place the picked-up object on the pre-applied droplet of (3);(8)The XY-stage is moved such that the object is within view of the CCD camera to measure the X- and Y- coordinates.

In steps (3), (5), and (7) of the above process, the amount of Z-stage movement is calculated and determined by the image processing program written in Python from images captured by the side camera.

## 5. Image Processing

In this section, we explain the vision feedback system. The program used Open CV by Python. The system was used when applying place droplets, picking up, and placing objects. Figure 6 shows the image of the process of the vision feedback system in placement, and this process is described as follows:
(1)The side camera captures the image so that both the object and the place droplets are visible (Figure 6a);(2)The captured image is imported to the PC and the Sobel filter is applied (Figure 6b).(3)Next, the image is binarized by the threshold (Figure 6c);(4)The image is then used to detect the contour of the object (Figure 6d);The distance $l_{1}$ between the object and the place surface is calculated from the coordinates of the lowest surface of the object (Figure 6e);(6)In this experiment, the parameter is the liquid bridge height *h*_2_ between the object and the place surface at placing (Figure 6f). Thus, from *h*_2_ and $l_{1}$, the stage descent amount $l_{2}$ is determined as follows:(9)$$l_{2}=l_{1}-h_{2}.$$(7)The amount of Z-stage descent determined in (6) is fed back to the control (Figure 6g).

As described in step 6, we used *h*_2_ as a parameter in this experiment. Here, Table 3 shows the set values of *h*_2_ for each experimental condition.

Here, the condition values of (C1)–(C5) are the same as in Table 2. We conducted the pick-and-place experiment 10 times for each condition.

## 6. Experiments

### 6.1. Comparison of Results with Height h_2_

We conducted the pick-and-place operation under the five conditions. Figure 7 shows the sequence of pick-and-place operations under the condition of (C3) for *h*_2_ of 50 μm.

In Table 4, the success rates and positioning errors for *h*_2_ = 0, 50, 70, 100, and 120 μm are compared. From Table 4 and Figure 7, the newly developed vision feedback system confirmed that the gripper could automatically pick-and-place objects without solid contact. Figure 8 shows the relationship between the positioning errors and *h*_2_ under the condition (C3).

The smallest value of the positioning error was 44 ± 34 μm with *h*_2_ = 70 μm (the ratios to the cube side were 4.4 ± 3.4%), although the error with *h*_2_ = 50 μm had almost the same value.

Figure 9 shows the typical examples for a large positioning error. The water spread to the sides of the object when *h*_2_ was too small, as shown in Figure 9a, whereas the cube was easily inclined when *h*_2_ was too large, leading to a large positioning error, as shown in Figure 9b.

### 6.2. Evaluation of Size Matching Effect

The positioning errors for each condition are compared in Table 5. We used the result with the smallest error out of the *h*_2_ values set for each condition.

The cube moves toward the center position of the droplet on the surface due to surface tension as a self-alignment phenomenon. If the average error is only attributed to the center position of the pre-applied droplet, the errors of (C1) and (C3) should be almost equal because the same nozzle is used for applying the droplet to the surface. For the same reason, that of (C2) and (C4) should also be almost equal. However, twice the difference between (C1) and (C3) and quadruple the difference between (C2) and (C4) can be observed from Table 5. We supposed that the size and shape matching effects also influenced the final alignment errors [29,30,31].

To examine the size matching effect, we defined the diameter of the circular nozzle as *D*, the positioning error as *E*, and the side length of the cube as *L*. We normalized the parameters (D, E) by *L* as follows:(10)$$D^{*}\equiv D/L,$$
(11)$$E^{*}\equiv E/L.$$

We compared the results in Table 5 using these parameters to verify a suitable size of circular shaped nozzle for each object, as shown in Figure 10.

As shown in Figure 10, the closer *D** is to 1, the smaller the positioning error becomes. When the nozzle diameter was approximately twice as large as the object as in condition (C1), the distribution of the final alignment position was larger than those of (C2) and (C3). When the nozzle diameter was approximately half of the object, as in condition (C4), the object was often tilted during pick-up, as shown in the photograph of Figure 10 (C4). That is the main reason why (C4) obtained the largest distribution among the four conditions.

### 6.3. Attitude Control Using Shape Matching Effect

We considered that the attitude angle of the picked-up object could be controlled by matching the shape of the nozzle to the picked object.

To examine the shape matching effect [30,31], we conducted the pick-and-place experiment as in condition (S) in Table 4 with a square-shaped nozzle and side length of 1 mm for the 1-mm cube. Figure 11 shows a sequential photograph of the pick-and-place operation.

From Figure 11a,b, the object changed its attitude angle for aligning the nozzle shape; we verified that the shape matching effect is useful for controlling the attitude of the sample. The movement of the object is illustrated in Figure 12.

We compared the experimental results with condition (C3) using a circular nozzle to conduct a pick-and-place of the same cubed object. The positioning error and the attitude angle of the object after placement are shown in Table 6, and the image of the object after placement is shown in Figure 13.

As no target angle was set in these experiments, we focused on the standard deviations (SDs) to discuss the experimental results of the accuracy of the attitude angle. Table 6 shows that the SD of the attitude angle under condition (S) was smaller than condition (C3). Using this nozzle and a rotational stage, we could set the objects at various angles, as shown in Figure 14.

## 7. Conclusions and Future Prospects

This paper described a newly developed vision feedback method for improving the placement accuracy for a single nozzle capillary gripper. The developed system was coded in Python using OpenCV.

In the experiment, we automatically picked-and-placed the cubes without any solid contact, and the success rate was 99% (129 out of 130 samples). For the cubes with 1- and 0.5-mm sides, the minimum positioning error ± SD of each size was 44 ± 34 μm and 39 ± 18 μm, respectively; the ratios of the positioning errors to the lengths of the objects were 4.4 ± 3.4% and 7.8 ± 3.6%, respectively. We also verified that the size matching effect worked reasonably; the average positioning error minimized when the size of the nozzle was nearly equal to the side of the cube.

In addition, using a square-shaped nozzle with the same shape of the cube, we succeeded in controlling the attitude; the SD of the attitude angle was reduced from a random value to 7.4°.

To reduce the positioning errors, our future plans are as follows. We plan to evaluate the relationship between the surface shape and properties of the objects as well as the radius and arrangement pattern of the droplets, for effectively applying the self-alignment phenomena. Further work will study the accuracy of contour detection in the vision feedback system. The impact of lighting on positioning will also be studied in the future, for example, the effects of droplet size and color and diffraction and refraction of lighting. In addition, the investigation of the release condition is a significant subject from both theoretical and experimental approaches.

Moreover, we wish to take on the challenge of developing a pick-and-place mechanism for smaller complex-shaped objects of less than 200 µm, such as micro-LEDs, helical-shaped coils, thin wires, gears, soft contact lenses, fragile gels, and biomedical cells with the capillary grippers to cultivate the manipulation of heterogeneous and complex-shaped micro-objects in flexible electronics, micro-electro-mechanical systems (MEMS), soft robotics, soft matter, and biomedical fields. In the microminiaturization of an object, we consider the limitations to be determined as follows: the smallest possible droplet size, the sharpest nozzle, the effect of droplet evaporation, and the piezoelectric control of droplet volume.

## Figures and Tables

**Figure 1 micromachines-13-01270-f001:**
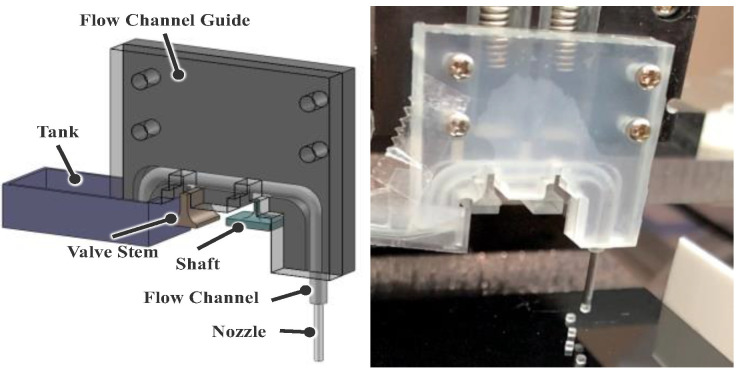
Working area of the pick-and-place of 1-mm cubes.

**Figure 2 micromachines-13-01270-f002:**
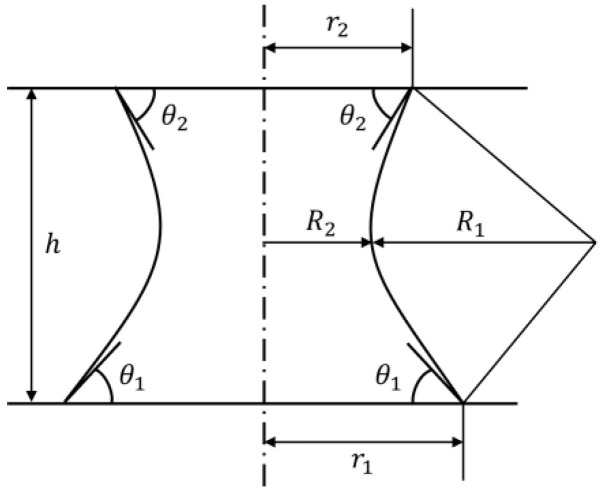
Capillary bridge between two parallel planes.

**Figure 3 micromachines-13-01270-f003:**
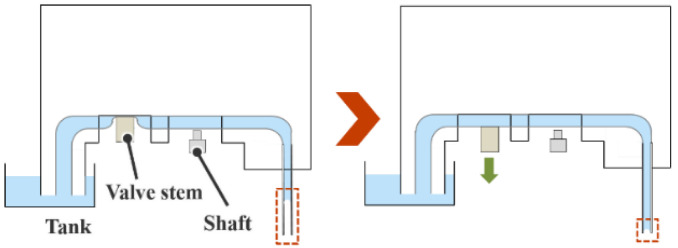
Mechanism of water filling till the tip of the channel by capillary action.

**Figure 4 micromachines-13-01270-f004:**
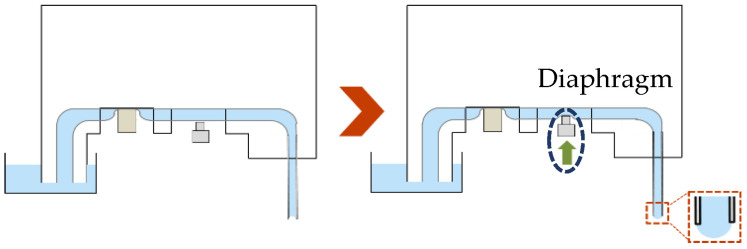
Principle of droplet formation.

**Figure 5 micromachines-13-01270-f005:**
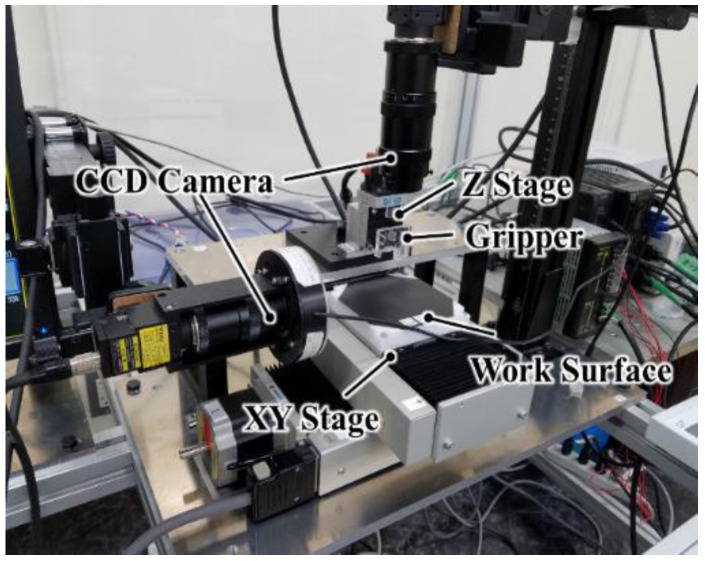
Experimental setup for the automatic pick-and-place.

**Figure 6 micromachines-13-01270-f006:**
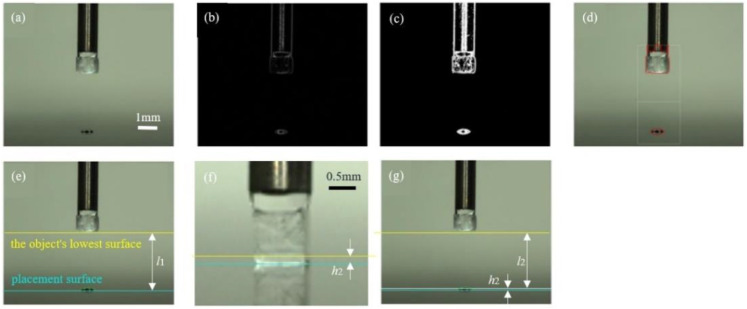
Process of the vision feedback system in placing: (**a**) the capture image, (**b**) the image with the Sobel filter, (**c**) the binarized image, (**d**) detecting contours, (**e**) calculating the distance *l*_1_, (**f**) the definition of *h*_2_, and (**g**) the definition of *l*_2_.

**Figure 7 micromachines-13-01270-f007:**
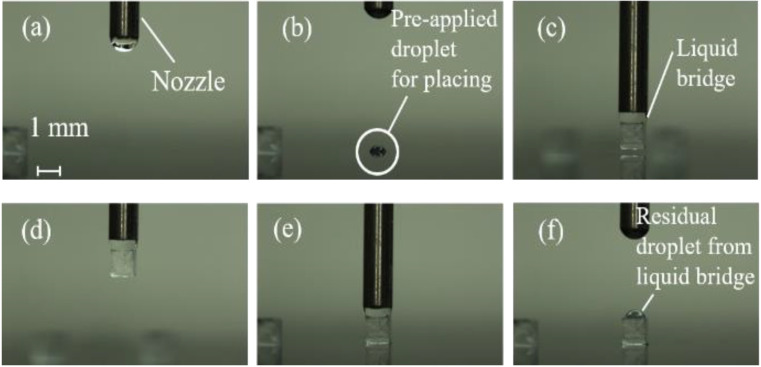
Sequential photograph of the pick-and-place experiment under (C3) for *h*_2_ = 50 μm: (**a**) droplet discharge from the nozzle, (**b**) application of the droplet by stamping the nozzle to the surface, (**c**) forming the liquid bridge between the nozzle and top surface of the cube, (**d**) picking up the cube, (**e**) placing the cube on the droplet on the substrate, and (**f**) releasing the cube by moving up the gripper.

**Figure 8 micromachines-13-01270-f008:**
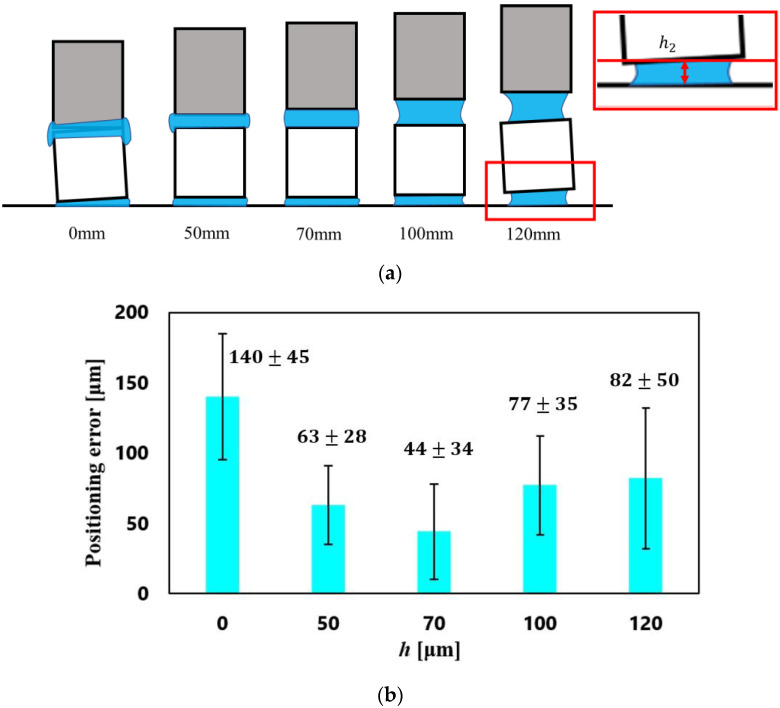
Relationship between the positioning errors and *h*_2_ under (C3). (**a**) Typical schematic diagram for each *h*_2_, (**b**) plots of positioning errors vs. *h*_2_.

**Figure 9 micromachines-13-01270-f009:**
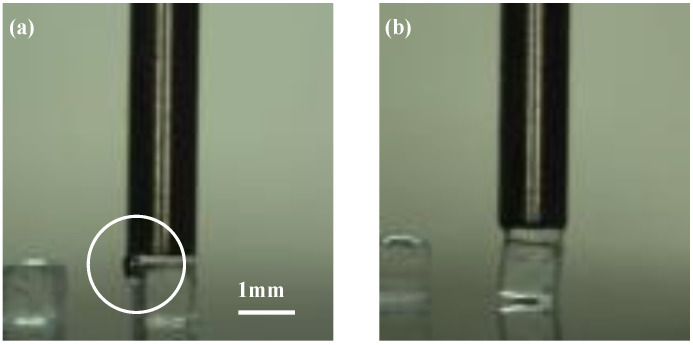
Example of a large positioning error: (**a**) *h*_2_ = 0 μm; (**b**) *h*_2_ = 100 μm.

**Figure 10 micromachines-13-01270-f010:**
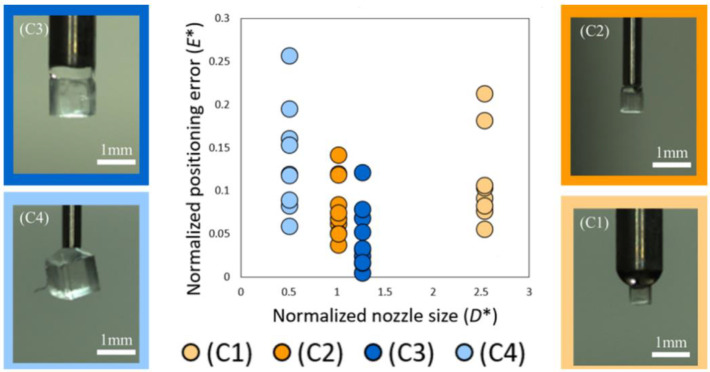
Plots of nozzle size *D** vs. positioning error *E**.

**Figure 11 micromachines-13-01270-f011:**
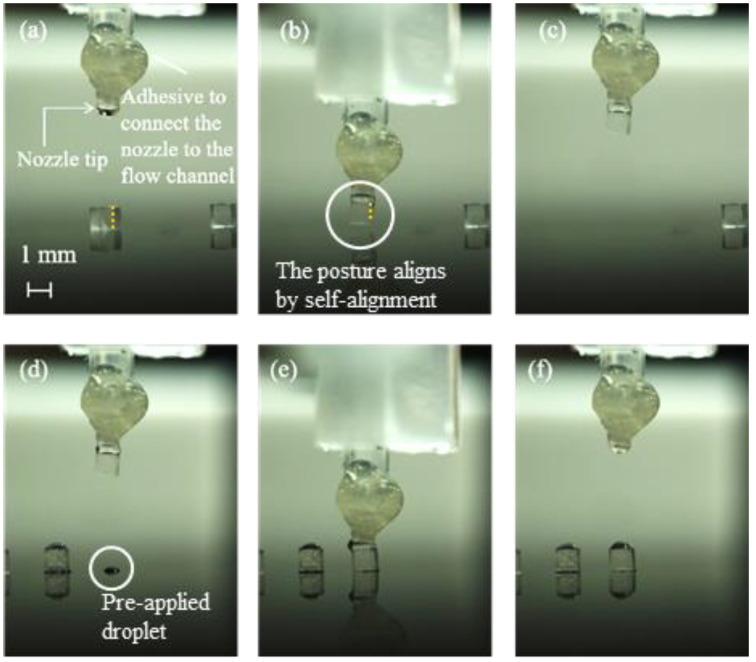
Sequential photograph of the pick-and-place of a 1-mm cube by the square nozzle: (**a**) droplet discharge from the nozzle, (**b**) forming the liquid bridge between the gripper and cube with a self-alignment of the attitude, (**c**) picking up the cube, (**d**) positioning the nozzle just above the pre-applied droplet, (**e**) placing the sample on the droplet on the substrate, and (**f**) releasing the sample by moving the gripper up.

**Figure 12 micromachines-13-01270-f012:**
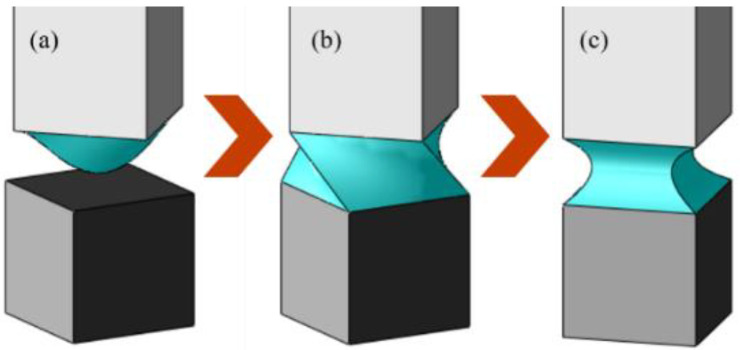
Shape matching sequence between same-sized square nozzle and cube: (**a**) droplet discharge from the square nozzle, (**b**) just before the liquid spreading and making the liquid bridge between two surfaces, and (**c**) liquid bridge deformed into a symmetrical shape such that the surface energy of the liquid bridge is minimized.

**Figure 13 micromachines-13-01270-f013:**
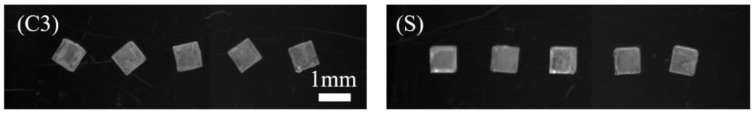
Arrangement of 1-mm cubes under conditions (C3) and (S).

**Figure 14 micromachines-13-01270-f014:**
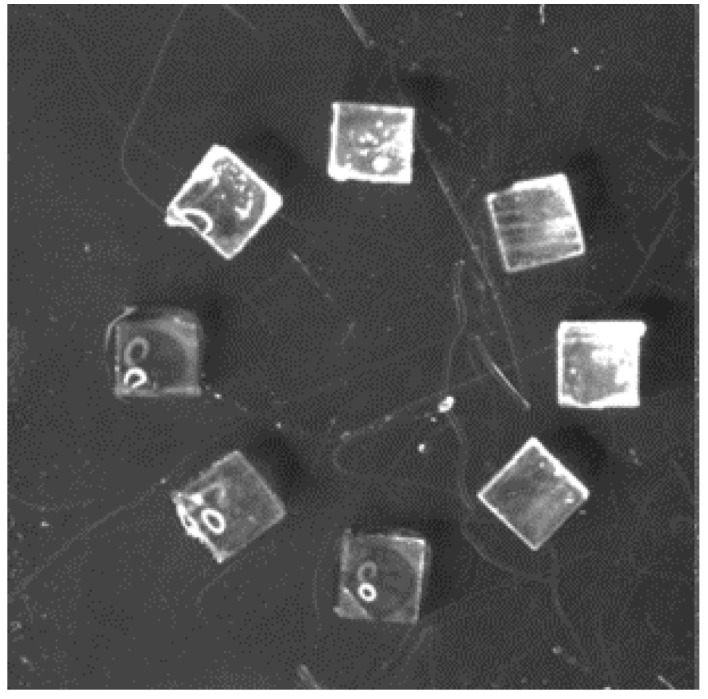
Circular arrangements of 1-mm cubes at various angles.

**Table 1 micromachines-13-01270-t001:** Specifications of micro-objects.

Name	1 mm Cube	0.5 mm Cube
Image	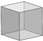	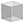
Material	Acrylic
Geometric parameters	Depth: 1 mmWidth: 1 mmHeight: 1 mm	Depth: 0.5 mmWidth: 0.5 mmHeight: 0.5 mm
Gravity force	12 μN	1.5 μN

**Table 2 micromachines-13-01270-t002:** Experimental condition and the parameters.

Condition	(C1)	(C2)	(C3)	(C4)	(S)
Picked Object	0.5 mm Cube	1 mm Cube
Nozzle’sscross section	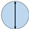	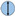	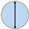	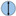	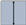
Parameters	Designparameters	r2	0.635 mm	0.255 mm	0.635 mm	0.255 mm	0.5 mm
Experimental parameters	γ	72.9 μN
h	0.52 mm	0.25 mm	0.35 mm	0.23 mm	0.50 mm
θ1	148.1°	100°	77.3°	73.6°	100.4°
θ2	73.7°	109°	82.1°	80°	86.8°
Estimatedcapillary force from (8)	97.6 μN	163 μN	256.5 μN	197 μN	260 μN
Gravity force	1.5 μN	12 μN

**Table 3 micromachines-13-01270-t003:** Targeted values of $h_{2}$.

Condition	(C1)	(C2)	(C3)	(C4)	(S)
Cube size	0.5 mm		1 mm		
Nozzle shape	Circle				Square
Nozzle size (mm)	1.27	0.51	1.27	0.51	1 × 1
	Diameter				
$h_{2}$ (μm)	10 25 50	10 25 50	0 50 70 100 120	25 50	70

**Table 4 micromachines-13-01270-t004:** Comparison of success rate and position errors.

$h_{2}\;(\mu m)$		0	50	70	100	120
Success rate	Pick-up (%)	100	100	100	100	100
	Place-down (%)	100	90	100	100	100
Positioning error	Average (µm)	140	63	44	77	82
	Standard deviation (µm)	45	28	34	35	50

**Table 5 micromachines-13-01270-t005:** Comparison of the positioning errors for conditions (C1), (C2), (C3), and (C4).

Condition	(C1)	(C2)	(C3)	(C4)
Ratio of nozzle size to object size	254 (%)	102 (%)	127 (%)	51 (%)
*h* _2_	25 (μm)	25 (μm)	70 (μm)	25 (μm)
Average error	80 (μm)	40 (μm)	44 (μm)	154 (μm)
Standard deviation	53 (μm)	17 (μm)	34 (μm)	77 (μm)
Ratio of error to object size	16 ± 10.6 (%)	8.0 ± 3.4 (%)	4.4 ± 3.4 (%)	15.4 ± 7.7 (%)

**Table 6 micromachines-13-01270-t006:** Comparison of the positioning errors and attitude angles for conditions (C3) and (S).

Condition		(C3)	(S)
Positioning error	Average	44 (μm)	99 (μm)
	Standard deviation	34 (μm)	57 (μm)
Attitude angle	Average	−3.0 (deg.)	−2.8 (deg.)
	Standard deviation	26.9 (deg.)	**7.4** (deg.)

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
