# Peer review of "Vision Feedback Control for the Automation of the Pick-and-Place of a Capillary Force Gripper"

_micromachines, 2022, doi:10.3390/mi13081270_

Round 1

Reviewer 1 Report

1)     In the manuscript, the term “image feedback” would be better replaced by vision feedback, which is more professional.

2)      The plots in Fig. 8 are blocked by the video snapshots?

3)     Fig. 10 is puzzling. What’s the relationship between C1~4 and the plot at the center?

4)     The format of the references is not consistent. For example, the page numbers for Ref [1] and [5] are not presented in the same format. Please go through the references and fix any issues.

5)     The language requires to be polished. For example, there are too many “we” in the manuscript.

Reviewer 2 Report

The following questions/issues should be addressed to carefully before resubmission:

- Table 2, left column: The titles "Design parameters" & "Experimental parameters" should be given in horizontal.

- What are the field of depth of the cameras used? What are the  traverse limits of the stage used in the experiments? 

- How does the resolution of the stage affects the accuracy and repeatibility of the described pick-and-place operation?

- What is the numerical aperture of the camera-lens combinations? How does the lighting affect the performance? What lightwaves are expected to perform better with different droplet size and colors? Do diffraction and refraction pose as a detrimental effect or does either have no effect on the overall operation when the contour detection is considered?

- Figure 8 seems to be somehow shadowed by figure 9. Please kindly submit a corrected version with the revised manuscript.

- Table 6, left hand side: The title "Position error" seems to have allignment problems.

- The technique and the application seems promising. There is but one issue with the overall system: In the section "conclusions and future prospects" authors stated their future plan to manipulate much smaller objects. Here, I believe, a small discussion on 'what the theoretical & practical limiting factors are' is warranted. The said discussion would increase the impact of the work greately. This, I believe, is also related to some of my earlier questions in this review - and it will shed a light on the measured errors reported in the manuscript.

- Final comment: How fast does the overall system operate? Is the control real-time? There should be a trade-off between the speed and accuracy. Can the authors provide a metric on that based on measurements? If not, still, a theoretical limit will be quite useful for the readers.

Reviewer 3 Report

The paper Image Feedback Control for Automation of Pick-and-Place of 2 Capillary Force Gripper is very interesting but the presentation of the covered topic should be improved a great deal.

Please, find all my comments and suggestions for your paper improvement attached.

Round 2

Reviewer 3 Report

Please, find attached my suggestions for improvements of your paper.
